# Proton-transporting heliorhodopsins from marine giant viruses

**Shoko Hososhima[1], Ritsu Mizutori[1], Rei Abe-Yoshizumi[1], Andrey Rozenberg[2], Shunta Shigemura[1], Alina Pushkarev[2†], Masae Konno[1‡], Kota Katayama[1,3], Keiichi Inoue[1‡], Satoshi P Tsunoda[1,3], Oded Béjà[2], Hideki Kandori[1,3]***

[1]Department of Life Science and Applied Chemistry, Nagoya Institute of Technology, Showa-ku, Japan; [2]Faculty of Biology, Technion-Israel Institute of Technology, Haifa, Israel; [3]OptoBioTechnology Research Center, Nagoya Institute of Technology, Showa-ku, Japan

**Abstract** Rhodopsins convert light into signals and energy in animals and microbes. Heliorhodopsins (HeRs), a recently discovered new rhodopsin family, are widely present in archaea, bacteria, unicellular eukaryotes, and giant viruses, but their function remains unknown. Here, we report that a viral HeR from Emiliania huxleyi virus 202 (V2HeR3) is a light-activated proton transporter. V2HeR3 absorbs blue-green light, and the active intermediate contains the deprotonated retinal Schiff base. Site-directed mutagenesis study revealed that E191 in TM6 constitutes the gate together with the retinal Schiff base. E205 and E215 form a PAG of the Schiff base, and mutations at these positions converted the protein into an outward proton pump. Three environmental viral HeRs from the same group as well as a more distantly related HeR exhibited similar proton-transport activity, indicating that HeR functions might be diverse similarly to type-1 microbial rhodopsins. Some strains of *E. huxleyi* contain one HeR that is related to the viral HeRs, while its viruses *Eh*V-201 and *Eh*V-202 contain two and three HeRs, respectively. Except for V2HeR3 from *Eh*V-202, none of these proteins exhibit ion transport activity. Thus, when expressed in the *E. huxleyi* cell membranes, only V2HeR3 has the potential to depolarize the host cells by light, possibly to overcome the host defense mechanisms or to prevent superinfection. The neuronal activity generated by V2HeR3 suggests that it can potentially be used as an optogenetic tool, similarly to type-1 microbial rhodopsins.

**\*For correspondence:**
kandori@nitech.ac.jp

**Present address:** [†]Institute of Biology, Experimental Biophysics, Humboldt-Universität zu Berlin, Berlin, Germany; [‡]The Institute for Solid State Physics, The University of Tokyo, Kashiwa, Japan

**Competing interest:** The authors declare that no competing interests exist.

## Editor's evaluation

This manuscript provides the first experimental evidence that some members of the newly discovered heliorhodopsins can function as proton channels. The authors provide evidence of this transport function as well as a characterization of the photo cycle. The authors also demonstrate that these heliorhodopsin proton channels can be utilized as optogenetic tools. These findings should be of interest to a wide audience interested in membrane biophysics as well as in the development of tools for neuroscience.

## Introduction

Many organisms perceive light using rhodopsins (*Ernst et al., 2014*; *Govorunova et al., 2017*; *Grote et al., 2014*; *Rozenberg et al., 2021*), integral membrane proteins containing retinal chromophores. Rhodopsins are classified into type-1 microbial and type-2 animal rhodopsins, which contain all-*trans* and 11-*cis* retinal, respectively. Type-2 animal rhodopsins function as G-protein-coupled receptors, while the molecular functions of type-1 microbial rhodopsins are highly diverse and include light-driven ion pumps, light-gated ion channels, light sensors, and light-activated enzymes. Ion-transporting

rhodopsins are used as the main tools to control membrane potential in optogenetics (*Deisseroth and Hegemann, 2017*). In addition to type-1 and type-2 rhodopsins, a previously unrecognized divergent family of heliorhodopsins (HeRs) was recently discovered using functional metagenomics (*Pushkarev et al., 2018*). HeRs are distant relatives of type-1 rhodopsins, and their structures and photocycles resemble those of other microbial rhodopsins (*Kovalev et al., 2020*; *Lu et al., 2020*; *Pushkarev et al., 2018*; *Shihoya et al., 2019*). The most unusual structural feature of HeRs is that their membrane topology is inverted compared to type-1 and -2 rhodopsins (*Pushkarev et al., 2018*; *Shihoya et al., 2019*).

Similarly to type-1 rhodopsins, HeRs are encoded in genomes of archaea, bacteria, unicellular eukaryotes, and giant viruses. Physiological functions of HeRs, however, remain unknown. Previous studies did not detect any ion transport activity in HeRs, and based on their slow photocycle, a sensory function was suggested (*Pushkarev et al., 2018*; *Shihoya et al., 2019*). Analogously, no ion-transport function is consistent with the crystal structures of the prokaryotic HeRs *Ta*HeR (from a *Thermoplasmatales* archaeon) (*Shihoya et al., 2019*) and 48C12 (from an actinobacterium) (*Kovalev et al., 2020*; *Lu et al., 2020*), in which the interior of the extracellular half is highly hydrophobic. It should be noted, however, that these structures provide a very limited representation of the family. HeRs are a highly diverse group (*Kovalev et al., 2020*; *Chazan et al., 2022*) for which a variety of functions might be expected. In the present study, we report ion transport activity for a viral HeR, which provides support to the idea of functional diversity among HeRs.

The HeRs studied here come from giant double-stranded DNA viruses from the genus *Coccolithovirus* (*Phycodnaviridae*) that infect the microalga *E. huxleyi* (=*Gephyrocapsa huxleyi*). *E. huxleyi* is a globally important marine coccolithophore whose massive blooms are observable from satellites and have an impact on Earth's climate (*Thierstein and Young, 2004*). *E. huxleyi* viruses are able to collapse its blooms and thus represent one of the main factors controlling abundance of *E. huxleyi* in the ocean (*Bidle et al., 2007*; *Vardi et al., 2009*). Curiously, coccolithoviruses encode HeRs in their genomes, with some isolates, such as *Eh*V-202, having up to three HeR genes (*Figure 1A*). One of the HeR genes from *Eh*V-202 (AET42570.1; V2HeR2) has been expressed before and failed to demonstrate any light-dependent ion-transporting activity (*Shihoya et al., 2019*). Despite this, here we report the detection of photocurrents for a different HeR from *Eh*V-202 as well as several other viral HeRs. We suggest a molecular mechanism for ion transport in these HeRs and discuss their evolution and physiological role.

## Results

Driven by the hypothesis that there might exist a functional diversity among the three HeRs in *Eh*V-202, we first targeted V2HeR3 (AET42421.1), a HeR with only ~30% identity to V2HeR2. We expressed it in a cultured mammalian cell line (ND7/23), visualized its expression by P2A-linked eGFP and a cMyc epitope tag and applied patch-clamp recordings in an attempt to detect photocurrents. Vector expression in ND7/23 cells was confirmed by observing eGFP signal on the cytoplasmic side (green in *Figure 1B*). Clear membrane fluorescence from anti-cMyc tag was observed only from the C-terminal cMyc, but not from the N-terminal cMyc (magenta in *Figure 1B*), indicating that the C-terminus faces the extracellular side, as in other HeRs (*Pushkarev et al., 2018*; *Shihoya et al., 2019*). Remarkably, the V3HeR-expressing ND7/23 cells yielded consistent photocurrent responses in the patch-clamp experiments (*Figure 1C*). The photocurrents exhibited a sharp negative transient peak ($I_0$), which rapidly dropped into a relatively broad peak component ($I_1$), followed by a steady-state current component ($I_2$) during the course of illumination. The presence of steady-state currents is an unequivocal indication of ion transport maintained for a long time span (*Figure 1—figure supplement 1*). Photocurrent amplitude increased with light intensity and demonstrated a single sigmoid curve (*Figure 1—figure supplement 2*), indicative of photocurrents owing to single-photon events by unphotolyzed molecules. Components $I_1$ and $I_2$ exhibited a linear relation to the voltage in the current-voltage (I-V) plot (see *Figure 1C*), with a reversal potential ($E_{rev}$) at +30 and +40 mV, respectively, while $I_0$ was always negatively directed and exhibits a weak voltage dependency. The passive currents of $I_1$ and $I_2$ imply a light-gated channel function (*Figure 1C*) and the linear I-V relationship suggests that V2HeR3 is a voltage-independent channel.

Next, we focused on the ion selectivity of V2HeR3. The I-V plot under the symmetric ionic conditions on both sides of the membrane without metal cations (pH 7.4) was almost linear with $E_{rev}$ of

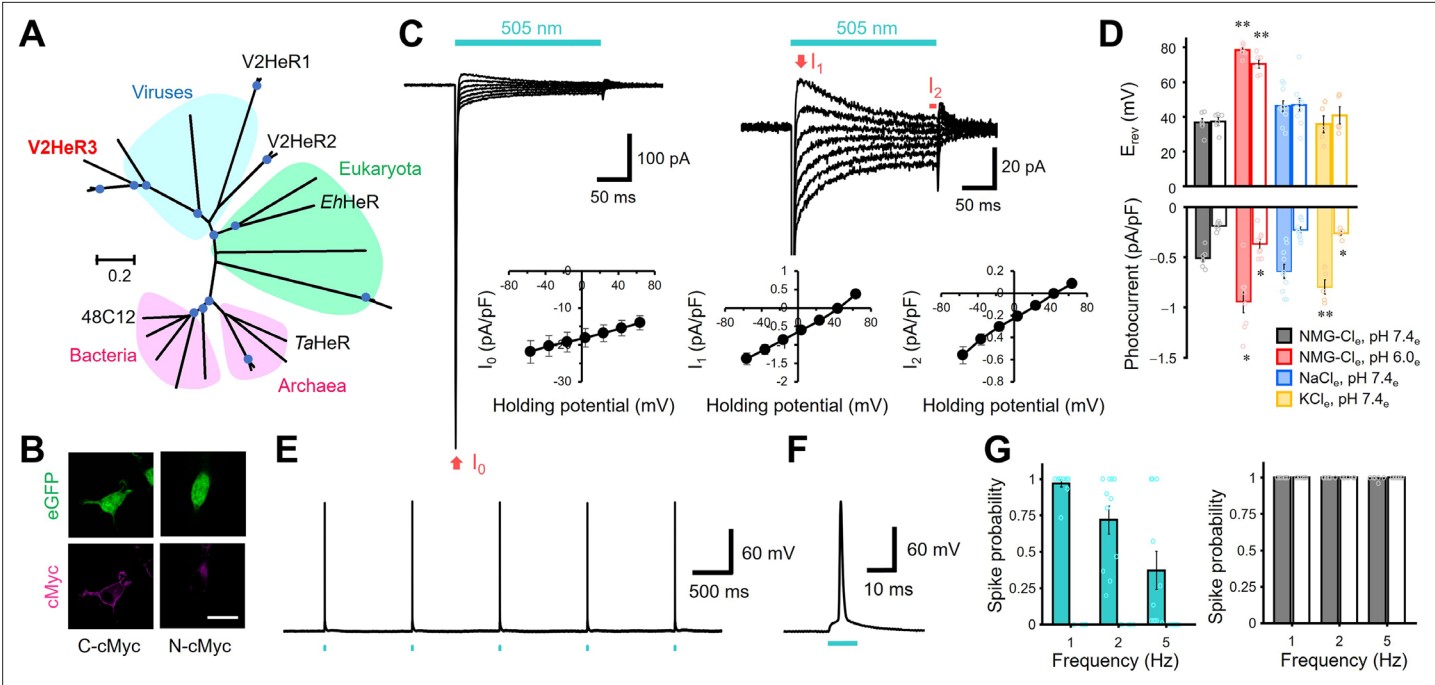

**Figure 1.** Light-gated inward proton transport of a viral heliorhodopsin (HeR) from Emiliania huxleyi virus 202 (V2HeR3). (**A**) Phylogenetic tree of HeRs, which includes three viral HeRs from *E. huxleyi* 202 (V2HeR1-3), a eukaryotic HeR from *E. huxleyi* (Ehux-HeR), an archaeal HeR (*Ta*HeR) and a bacterial HeR (48C12). (**B**) eGFP fluorescence (top, green) and immunofluorescence staining (bottom, magenta) observation of V2HeR3 with a cMyc epitope tag at the C terminus (left) and the N terminus (right) in cultured ND7/23 cells. Scale bar, 20 µm. (**C**) Electrophysiological measurements of V2HeR3-driven photocurrent in ND7/23 cells. The cells were illuminated with light ($\lambda$ =505 nm, 24.5 mW/mm$^2$) during the time region shown by the blue bars. The membrane voltage was clamped from −60 to +60 mV for every 20 mV step. The pipette solution was 110 mM NMG-Cl$_i$, pH 7.4$_i$, the bath solution was 140 mM NaCl$_e$, pH 7.4$_i$n=10 cells. (**D**) Corresponding reversal voltage (E$_{rev}$) for each internal condition (upper), and comparison of photocurrent amplitudes at 0 mV for different internal cations (bottom). Square-block bar graph indicates E$_{rev}$ or amplitude from peak photocurrent (I$_1$), open bar graphs indicate E$_{rev}$ or amplitude from steady-state photocurrent (I$_2$). The pipette solution was 110 mM NMG-Cl$_i$, pH 7.4$_i$, the bath solution was 140 mM NMG-Cl$_e$, pH 7.4$_e$ (black), 140 mM NMG-Cl$_e$, pH 6.0$_e$ (red), 140 mM NaCl$_e$, pH 7.4$_e$ (blue) or 140 mM KCl$_e$, pH 7.4$_e$ (yellow). n=5–10 cells. (*p<0.05, **p<0.01). (**E**) Representative responses of a V2HeR3-expressing neuron to 10 ms light pulses (left, $\lambda$ =505 nm, 24.5 mW/mm$^2$) at 1 Hz. (**F**) The firstaction potential in E. the X axis is expanded. (**G**) Comparison of spike probability by electrical stimulation (right, 300 pA current injections) or light stimulation (left, $\lambda$ =505 nm, 24.5 mW/mm$^2$). The Square-block bar indicates spike probability from V2HeR3-expressing neurons, the open bar indicates spike probability from the neurons without V2HeR3. n=6–11 cells.

The online version of this article includes the following figure supplement(s) for figure 1:

**Figure supplement 1.** Photocurrents of viral HeR from Emiliania huxleyi virus 202 (V2HeR3) under different illumination periods.

**Figure supplement 2.** Photocurrents of viral HeR from Emiliania huxleyi virus 202 (V2HeR3) under different light intensities.

**Figure supplement 3.** Selectivity of cations in the transport of viral HeR from Emiliania huxleyi virus 202 (V2HeR3).

**Figure supplement 4.** Effect of anions in the transport of viral HeR from Emiliania huxleyi virus 202 (V2HeR3).

**Figure supplement 5.** Ion transport activity of viral HeR from Emiliania huxleyi virus 202 (V2HeR3) in *Pichia pastoris* measured by a pH-electrode.

**Figure supplement 6.** Effect of anions in the I$_0$ component of viral HeR from Emiliania huxleyi virus 202 (V2HeR3).

**Figure supplement 7.** Kinetics properties of the I$_0$ component.

**Figure supplement 8.** Light power dependency of neuronal manipulation by viral HeR from Emiliania huxleyi virus 202 (V2HeR3).

about +40 mV (***Figure 1C***, ***Figure 1—figure supplement 3***). Lowering the extracellular pH from 7.4 to 6.0 resulted in an E$_{rev}$ shift from +40 to +70 mV (***Figure 1D***), while a shift from +40 to -6 mV was observed when pH$_i$ was lowered (***Figure 1—figure supplement 3***). By contrast, replacing the solutions with Na$^+$ or K$^+$ did not show any significant E$_{rev}$ shift (***Figure 1D***), suggesting that V2HeR3 is a light-gated proton channel. There was nevertheless a statistically significant difference in the current amplitude in the presence of K$^+$ (***Figure 1D***). This indicates that the H$^+$ transport is somehow enhanced in the presence of K$^+$ at the extracellular side.

One might argue that, if V2HeR3 functions as a pure channel, the photocurrent is expected to reverse at 0 mV under symmetric conditions. It should be noted in this respect that channelrhodopsin 2 from *Chlamydomonas reinhardtii* (ChR2) (*Nagel et al., 2003*), a standard depolarization tool in optogenetics (*Boyden et al., 2005*; *Deisseroth and Hegemann, 2017*; *Ishizuka et al., 2006*), possesses an outward proton pump activity and is thus a leaky proton pump (*Feldbauer et al., 2009*). Similarly, at low extracellular pH, the outward proton flux of the proton pump rhodopsin from *Gloeobacter violaceus* (GR) changes to a passive influx (*Vogt et al., 2013*). In the case of V2HeR3, significant positive $E_{rev}$ suggests that V2HeR3 also possesses the activity of a proton pump, although the direction is inward. The $E_{rev}$ shift upon pH change is smaller than that expected from Nernst potential (*Figure 1D*). Lowering the extracellular pH from 7.4 to 6.0 ($\Delta pH = 1.4$) is expected to correspond to a shift of 82 mV, yet only a 40–60 mV shift was observed for V2HeR3 (*Figure 1D* and *Figure 1—figure supplement 3B*). We thus conclude that the ionic current of V2HeR3 includes an $H^+$-pump component which is responsible for the deviation from the Nernst equation.

The current amplitude and the $E_{rev}$ were also altered depending on the monovalent anions in the solution (*Figure 1—figure supplement 4*). The inward current ($I_1$ and $I_2$) was suppressed in the presence of $Cl^-$, $Br^-$, and $NO_3^-$ in the bath solution (*Figure 1—figure supplement 4B* right) and the $E_{rev}$ values shifted accordingly (*Figure 1—figure supplement 4B* left). On the other hand, no significant change either in the current amplitude or in the $E_{rev}$ was observed when $Asp^-$ in the pipette solution was replaced with $Cl^-$ or $SO4^{2-}$ (*Figure 1—figure supplement 4C* right). Our interpretation of these results is that V2HeR3 contains a binding site for a monovalent anion at the extracellular side, and anion binding affects proton transport. Summarizing these electrophysiological experiments, we conclude that V2HeR3 functions as a proton channel with active inward $H^+$ transport. Further support to the idea that V2HeR3 conducts only protons was obtained using an ion transport assay with a pH electrode on V2HeR3 expressed in yeast (*Pichia pastoris*). Illumination of the V2HeR3-expressing yeast cells increased solution pH, which was abolished by the addition of the proton uncoupler CCCP (*Figure 1—figure supplement 5*).

Having established the origin of $I_1$ and $I_2$, we directed our attention to the initial sharp peak component $I_0$. The fast inward-directed current $I_0$ was significantly larger than $I_1$ and $I_2$ and the current-voltage relationship of $I_0$ was dramatically shifted to positive values with respect to the reversal potential (*Figure 1C1–V* plot, left). Extrapolation of the voltage dependence of $I_0$ predicted its reversal potential as +290 mV, indicating that $I_0$ is due to an ion pump component. No significant change was observed when $NMG^+$ in the extracellular solution was replaced with $Na^+$, or when $pH_e$ was lowered from 7.4 to 6.0 (*Figure 1—figure supplement 6A*). However, a substantial reduction in the current amplitude was observed when $pH_i$ was lowered to 6.0, but it was not affected by replacement of $Na^+$ as the intracellular cation (*Figure 1—figure supplement 6A*). Further, no significant differences in the I-V plot were observed when comparing $Asp^-$, $Cl^-$, and $SO_4^{2-}$ (*Figure 1—figure supplement 6C and D*), suggesting that anions do not contribute to $I_0$. The photocurrent of the component $I_0$ was expanded in *Figure 1—figure supplement 7*. The component $I_0$ reached its negative peak at 0.85 ms, independently from the membrane voltage (*Figure 1—figure supplement 7A and B*). The peak exhibited exponential relaxation with two time constants ($\tau_{off}$) (*Figure 1—figure supplement 7C*). Taken together, these observations could be interpreted as evidence that the inward-directed $H^+$ transport and $H^+$ exclusion are suppressed by high proton concentrations (pH 6.0) at the cytoplasmic side. In spite of the detailed experiments on the $I_0$ component above, it is difficult to distinguish ionic current from an intramolecular charge displacement. In fact, several studies proposed that the fast transient peak current in channelrhodopsins and $H^+$ pump rhodopsins reflects intramolecular proton transfer from the Schiff base to a proton acceptor residue (*Geibel et al., 2001*; *Sineshchekov et al., 2013*). Further experiments would thus be needed to clarify the origin of $I_0$.

To test the applicability of V2HeR3 for optogenetic manipulation of neuronal activity, we expressed V2HeR3 in cultured cortical neurons. Short light pulses at 505 nm repetitively triggered action potentials in the transfected cells at 1 Hz (*Figure 1E and F*). Spike probability approached 1.0 at 1 Hz pulse frequency whereas it decreased at 2 and 5 Hz (*Figure 1G*). The light power required for inducing the action potential was between 2 and 16 $mW/mm^2$, in the same range as required for ChR2 (*Figure 1—figure supplement 8A* and *Figure 1—figure supplement 8B*; *Boyden et al., 2005*; *Ishizuka et al., 2006*). Latency to spike peak equalled 4.7±0.60 ms at 25 $mW/mm^2$, but varied with varying light intensity (*Figure 1—figure supplement 8C*). The neurons remained excitable by current injections at

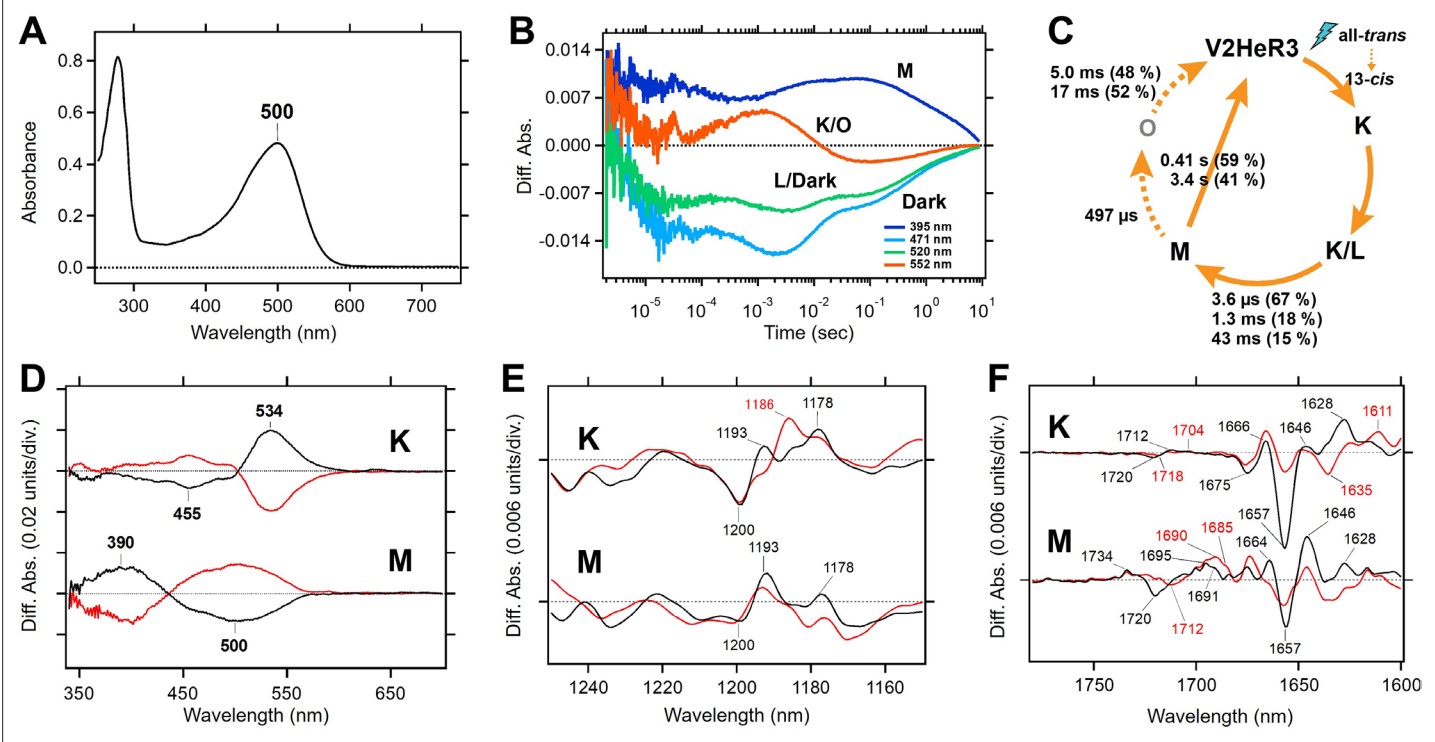

**Figure 2.** Molecular properties of the purified viral HeR from Emiliania huxleyi virus 202 (V2HeR3) proteins expressed in *Pichia pastoris* cells. (**A**) UV-visible absorption spectrum of V2HeR3 in detergent (0.1% *n*-dodecyl-β-D-maltoside [DDM]). (**B**) Time evolutions of transient absorption changes at characteristic wavelengths of specific photointermediates of V2HeR3. (**C**) Photocycle of V2HeR3 determined by analyzing the time evolution with multiexponential functions. (**D**) Light-induced low-temperature K-minus-dark (top) and M-minus-dark (bottom) difference UV-visible spectra of V2HeR3 obtained at 100 and 230K, respectively. Black curves represent the formation of the K and M intermediates by illuminating at 500 and>490nm, respectively, while red curves represent the reversion from the intermediates by illuminating at>530 and 400nm, respectively. (**E**) Light-induced low-temperature K-minus-dark (top) and M-minus-dark (bottom) difference FTIR spectra of V2HeR3 obtained at 100 and 230K, respectively, in the 1250–1150 cm$^{-1}$ region. (**F**) Light-induced low-temperature K-minus-dark (top) and M-minus-dark (bottom) difference FTIR spectra of V2HeR3 in H$_2$O (black) and D$_2$O (red) obtained at 100 and 230K, respectively, in the 1780–1600 cm$^{-1}$ region.

The online version of this article includes the following figure supplement(s) for figure 2:

**Figure supplement 1.** HPLC pattern of the retinal chromophore in viral HeR from Emiliania huxleyi virus 202 (V2HeR3).

**Figure supplement 2.** pH titration of viral HeR from Emiliania huxleyi virus 202 (V2HeR3).

**Figure supplement 3.** Light-induced difference FTIR spectra of viral HeR from Emiliania huxleyi virus 202 (V2HeR3) and *Ta*HeR at 77K.

**Figure supplement 4.** Light-induced difference FTIR spectra for late intermediates of viral HeR from Emiliania huxleyi virus 202 (V2HeR3) and *Ta*HeR.

each frequency even after the V2HeR3 expression, confirming that V2HeR3 itself does not harm the neurons (*Figure 1H*). These results demonstrate that V2HeR3 can be used for optical manipulation of neuronal excitability, although the frequency is limited to up to ca. 1 Hz.

Molecular properties of V2HeR3 were studied for the purified protein heterologously expressed in *P. pastoris* cells. The purified sample demonstrated an absorption maximum at 500 nm (*Figure 2A*). HPLC analysis revealed that V2HeR3 contains 64% all-*trans* retinal in the dark, which was converted to the 13-*cis* form by light (*Figure 2—figure supplement 1*). The p*K*$_a$ of the Schiff base and its counterion were determined to be 14.9 and 4.3, respectively (*Figure 2—figure supplement 2*), close to those of *Ta*HeR and 48C12 (*Pushkarev et al., 2018*; *Shihoya et al., 2019*). The photocycle was found to comprise a series of photointermediates: the primary red-shifted K intermediate, followed by the L intermediate, the deprotonated M intermediate, and the reprotonated O intermediate (*Figure 1C*). Although the photocycle upon illumination was slow as observed for other HeRs, the M intermediate in V2HeR3 was long-lived and directly returned to the original state (*Figure 2B*). Formation of the M intermediate possessed μs and ms components, suggesting that the M intermediate is the conducting state of the light-gated proton channel (*Figure 2C*). We then studied photointermediate states at low temperatures. *Figure 2D* shows formations of the K and M intermediates at 100

and 230 K, respectively, and their photoequilibria with the unphotolyzed state. FTIR analysis at 77 K (*Figure 2—figure supplement 3*) revealed a peak pair at 1200 (−)/1193 (+) cm$^{-1}$, characteristic of the all-*trans* to 13-*cis* photoisomerization (*Figure 2E*). Similar bands were observed for the M intermediate (*Figure 2E* and *Figure 2—figure supplement 4*), although the M intermediate generally lacked positive signals. This may originate from (1) vibration sources other than the retinal, (2) protonated photointermediates, or (3) photointermediates of the 13-*cis* photocycle. The negative peak at 1657 cm$^{-1}$ in *Figure 2F* is the characteristic vibration of helical amide-I, indicating that a structural perturbation α-helix takes place upon retinal isomerization (in the K intermediate), which is maintained in the M intermediate. Nevertheless, different structural changes in the protein are suggested by the stronger positive peaks at 1628 and 1646 cm$^{-1}$ for K and M intermediates, respectively. A peak pair at 1720 (−)/1712 (+) cm$^{-1}$ is indicative of a hydrogen-bonding change in a protonated carboxylic acid upon retinal isomerization, while the M intermediate shows peaks at 1734 (+)/1720 (−)/1695 (+) cm$^{-1}$ (*Figure 2F*). *Ta*HeR and 48C12 show no spectral changes in this frequency region (*Pushkarev et al., 2018*; *Shihoya et al., 2019*), and thus a protonated carboxylic acid of the C=O stretch at 1720 cm$^{-1}$ is unique for V2HeR3.

The above evidence strongly indicates that the mechanism of ion transport in V2HeR3 is different from other ion-transporting rhodopsins. In order to shed light on this mechanism, we characterized a series of V2HeR3 mutants by the use of site-directed mutagenesis. It is well known that internal carboxylates play an important role in such ion-transporting rhodopsins as ChR2 and bacteriorhodopsin

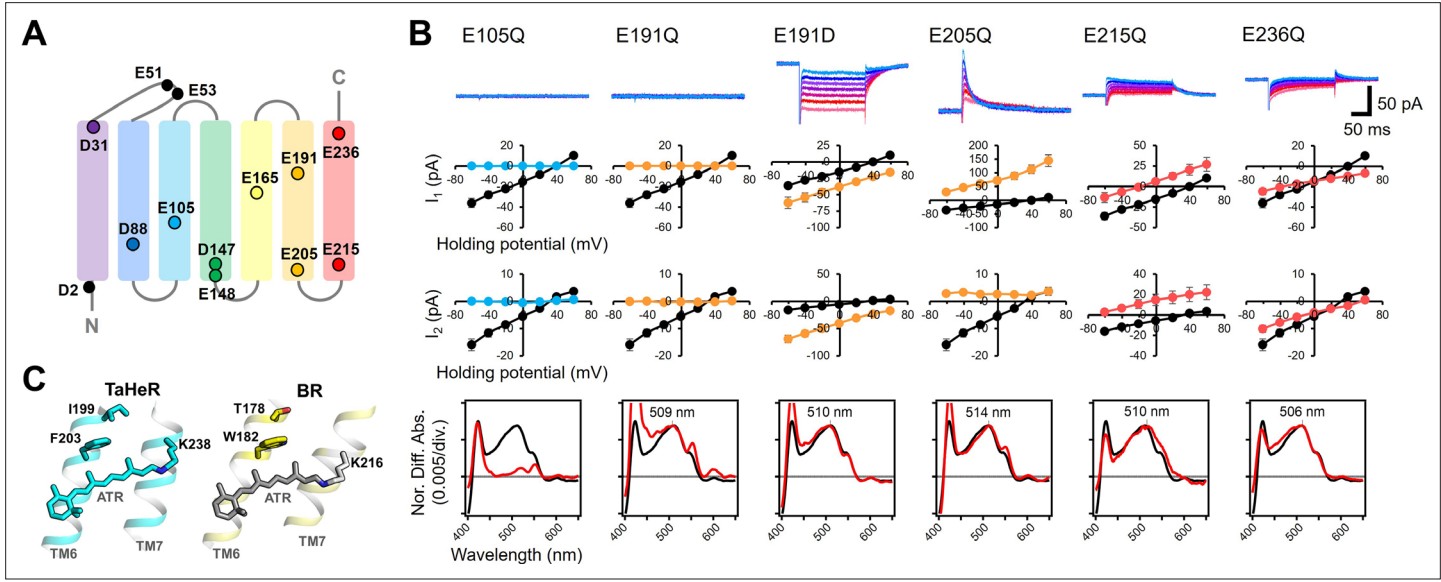

**Figure 3.** Electrophysiological analysis of the carboxylate mutants of viral HeR from Emiliania huxleyi virus 202 (V2HeR3). (**A**) Aspartate (**D**) or glutamate (**E**) in V2HeR3. Among 13 residues, 10 transmembrane aspartate and glutamate were replaced with asparagine (**D-to-N**) and glutamine (**E–to–Q**), respectively. (**B**) Photocurrents (top), I-V plots (middle), and absorption spectra (bottom) of mutants which differ from those of the wild type. Black curves in I-V plots and absorption spectra are the results of the wild type. The pipette solution was 110mM NaCl$_i$, pH 7.4$_i$, the bath solution was 140mM NaCl$_e$, pH 7.4$_e$. n=6–7cells. (**C**) Crystal structures of *Ta*HeR (left) and a light-driven proton pump bacteriorhodopsin (BR) (right). Corresponding residues of E191 in V2HeR3 are I199 in *Ta*HeR and T178 in BR. In TM6, highly conserved phenylalanine and tryptophan exist in HeRs and type-1 rhodopsins, respectively, while V2HeR3 contains tryptophan at this position.

The online version of this article includes the following figure supplement(s) for figure 3:

**Figure supplement 1.** Representative photocurrent traces of the wild-type and mutants of viral HeR from Emiliania huxleyi virus 202 (V2HeR3).

**Figure supplement 2.** Photocurrent-voltage plots (I-V plot) of the wild-type and mutants of viral HeR from Emiliania huxleyi virus 202 (V2HeR3).

**Figure supplement 3.** Absolute and difference absorption spectra of viral HeR from Emiliania huxleyi virus 202 (V2HeR3) in the experiments of hydroxylamine bleach.

**Figure supplement 4.** Absorption spectra of the wild-type (WT) and mutants of viral HeR from Emiliania huxleyi virus 202 (V2HeR3) obtained by the hydroxylamine bleach of ND7/23cells.

**Figure supplement 5.** Absorption spectra of the wild-type (WT) and mutants of viral HeR from Emiliania huxleyi virus 202 (V2HeR3) obtained by the hydroxylamine bleach of *Pichia pastoris* cells.

(BR) (*Gerwert et al., 2014*; *Kandori, 2020*; *Lórenz-Fonfría and Heberle, 2014*). V2HeR3 contains 13 carboxylates, among which D2 and E51/E53 are located at the N-terminus and extracellular loop, respectively, excluding the possibility that these residues are involved in the proton transport (*Figure 3A*). We thus prepared 10 mutants affecting the remaining carboxylate positions in all seven helices, and the photocurrents of the D-to-N or E-to-Q mutants were measured. *Figure 3—figure supplements 1 and 2* show the photocurrents and their I-V plots, respectively, measured by expressing each mutant in ND7/23 cells. Absorption spectra of each mutant in ND7/23 and *P. pastoris* cells were obtained without purification using the hydroxylamine bleaching method (*Figure 3—figure supplements 3–5*). Among the 10 mutants, those of carboxylates on TM1, TM2, TM4, and TM5 had little effect on the photocurrents, whereas mutations of E105, E191, E205, E215, and E236 on TM3, TM6, and TM7 did have a detectable impact (*Figure 3—figure supplement 1*). E105Q and E191Q in particular abolished the photocurrent entirely. E105 is the Schiff base counterion, and its neutralization led to loss of visible absorption (*Figure 3B* bottom). This was not the case for 48C12, as the corresponding mutation exhibited visible absorption upon binding chloride (*Singh et al., 2019*). In the case of E191Q, visible absorption remained but without photocurrent. The importance of carboxylate at position 191 is demonstrated further by the fact that the E191D mutant retained ion-transporting activity. Nevertheless, the I-V plots suggest that the E191D mutation diminished the proton channel mode converting the protein entirely to an inward proton pump, as indicated by the negative photocurrents. Positions homologous to E191 are I199 and T178 in *Ta*HeR and BR, respectively, which sandwich residues F203 and W182, respectively, with the retinal (*Figure 3C*). It should be noted that W182 exhibits unique conformational changes important for the proton pumping activity in BR (*Nango et al., 2016*; *Subramaniam and Henderson, 2000*; *Weinert et al., 2019*), and that the highly conserved tryptophan in type-1 microbial rhodopsins is replaced with phenylalanine in most HeRs. Interestingly, the proton-transporting V2HeR3 contains tryptophan at this position, which possibly constitutes the gate for the ion transport together with E191. In *Figure 3B*, E205Q and E215Q showed positive photocurrent signals at all the membrane voltages tested (−60 to +60 mV), suggesting proton conduction in the opposite direction (outward proton pump). It is thus likely that E205 and E215 are the key residues in defining the direction of proton transport. The current shape and the I-V plot of E236Q are similar to those of the wild type, although a shift of $E_{rev}$ to the more positive voltage from +30 to +60 mV is observed.

Despite significant sequence divergence (see *Supplementary file 3*), HeRs from coccolithoviruses form a monophylum close to HeRs from haptophytes and other algae, including a HeR gene found in *E. huxleyi* (*Figure 4—figure supplement 1*). Thus, in an attempt to trace the origins of the ion-transporting activity of V2HeR3, we investigated other viral HeRs, as well as the HeR gene from the host species. In addition to V2HeR3, *Eh*V-202 possesses HeRs from two more distantly related clades, V2HeR1 and V2HeR2, that alone are more widespread among *Eh*V isolates (*Figure 4—figure supplement 2*). Nevertheless, none of the four tested HeRs from clades V2HeR1 and V2HeR2 exhibited steady-state photocurrents as demonstrated in *Figure 4A* and *Figure 4—figure supplement 3*. A similarly negative result was obtained for *Eh*HeR, the HeR from the host alga. We then studied a collection of metagenomic viral HeR genes that were closely related to V2HeR3, as well as the more distantly related HeR from *Eh*V-PS401. *Figure 4B* and *Figure 4—figure supplement 4* clearly show that all of them exhibit ion transport activity similar to that of V2HeR3. The I-V plots of VPS401HeR and V*Tara*8957HeR show that the $E_{rev}$ is close to 0 mV, indicating that the proton channel mode is dominant (*Figure 4B*). In contrast, ion-transport properties of V*Tara*5482HeR and V*Tara*4616HeR are very similar to that of V2HeR3 judging from their I-V plots, indicating that the proton pump mode is prominent. The absorption spectra of these proteins obtained by hydroxylamine bleaching are shown in *Figure 4—figure supplement 5*. A sequence comparison of the HeRs is shown in *Figure 4—figure supplement 6*, with the key amino acids shown in *Figure 4C*. E105 of V2HeR3 is the Schiff base counterion, and E215 is conserved among HeRs. Among the other carboxylate residues, E205 and E236 are not fully conserved among ion-transporting HeRs, while E191 appears to be their hallmark. Interestingly, although relatively uncommon among eukaryotic HeRs, *Eh*HeR that did not demonstrate ion transport, as well as the more distantly related *Mc*HeR that was tested before (*Shihoya et al., 2019*), also contain the conserved glutamate E191 (see *Figure 4C* and *Figure 4—figure supplement 1*). This indicates that E191 might be essential but not sufficient for ion transport, and that other residues, such as W195 and E/Q205, are required as well.

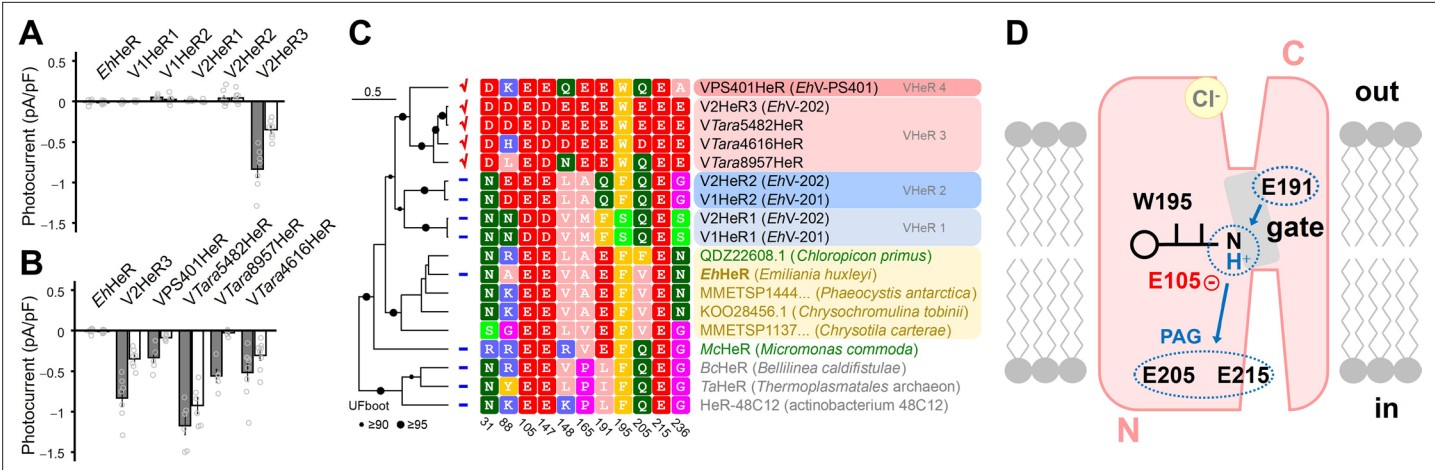

**Figure 4.** Electrophysiological measurements of related proteins of viral HeR from Emiliania huxleyi virus 202 (V2HeR3) and a functional model. (**A**) Electrophysiological measurements of a heliorhodopsin (HeR) from the host *Emiliania huxleyi* (Ehux), two HeRs from *E. huxleyi* virus 201 (V1HeR1, 2) and three HeRs from *E. huxleyi* virus 202 (V2HeR1-3). Although some HeRs exhibit transient photocurrents (positive or negative peaks), steady-state photocurrent was only observed for V2HeR3. Comparison of photocurrent densities at −40mV. Square-block bar indicates amplitude from peak photocurrent ($I_1$), open bar indicates amplitude from steady-state photocurrent ($I_2$). The pipette solution was 110mM $NaCl_i$, pH $7.4_i$, the bath solution was 140mM $NaCl_e$, pH $7.4_e$. n=5–8cells. (**B**) Electrophysiological measurements and the obtained I-V plots of homologous proteins of V2HeR3. Comparison of photocurrent amplitudes at −40mV. Square-block bar indicates amplitude from peak photocurrent ($I_1$), open bar indicates amplitude from steady-state photocurrent ($I_2$). The pipette solution was 110mM $NaCl_i$, pH $7.4_i$, the bath solution was 140mM $NaCl_e$, pH $7.4_e$. n=5–8cells. (**C**) Key residues for ion transport of HeRs. (**D**) Schematic drawing of suggested proton-transporting mechanism in V2HeR3.

The online version of this article includes the following figure supplement(s) for figure 4:

**Figure supplement 1.** Phylogenetic relationships of the *Eh*V heliorhodopsins (HeRs) and their eukaryotic cognates.

**Figure supplement 2.** Evolution of coccolithoviruses and their heliorhodopsin (HeR) genes.

**Figure supplement 3.** Representative photocurrent traces of heliorhodopsins (HeRs) from *Emiliania huxleyi* and its viruses.

**Figure supplement 4.** Representative photocurrent traces of viral HeR from Emiliania huxleyi virus 202 (V2HeR3)-like heliorhodopsins (HeRs).

**Figure supplement 5.** Absorption spectra of heliorhodopsins (HeRs) from *Emiliania huxleyi* and its viruses, and viral HeR from Emiliania huxleyi virus 202 (V2HeR3)-like HeRs obtained by the hydroxylamine bleach of ND7/23cells.

**Figure supplement 6.** Alignment of heliorhodopsins (HeRs).

## Discussion

*Figure 4D* outlines the proposed mechanism of proton channeling and pumping among the viral HeRs. Inward proton pumps were recently found in nature (*Harris et al., 2020*; *Inoue et al., 2020*; *Inoue et al., 2016*; *Shevchenko et al., 2017*), but the mechanism should be compared with that of outward proton pumps like BR, as the membrane topology in HeRS is opposite to that of type-1 and -2 rhodopsins (*Figure 1B*). Crystal structures of *Ta*HeR and 48C12 showed water-containing hydrogen-bonding networks between the retinal Schiff base and cytoplasmic aqueous phase. While the interior of the extracellular half is entirely hydrophobic in *Ta*HeR (*Shihoya et al., 2019*) and 48C12 (*Kovalev et al., 2020*; *Lu et al., 2020*), the proton-transporting HeRs reported here contain polar residues in that region. In particular, E191 appears to be the key residue for the function, presumably constituting the channel gate together with W195. Although E191 is a prerequisite for ion transport, it does not mean that this residue is negatively charged. Rather, E191 is protonated in the unphotolyzed state, and its deprotonation occurs upon opening of the gate. This residue is analogous to D96 in TM3 of BR and E90 in TM2 of ChR2, both of which are protonated in the dark state with the transient deprotonation necessary for light-driven proton pumping and light-gated cation channeling activities, respectively (*Braiman et al., 1988*; *Gerwert et al., 1989*; *Kuhne et al., 2019*; *Radu et al., 2009*). Similarly, in the cation channelrhodopsin 2 from *Guillardia theta* (GtCCR2) that shares with BR the DTD motif in TM3, channel opening requires deprotonation of the D96 homolog (*Sineshchekov et al., 2017*). Mutation of E205 or E215 in V2HeR3 led to conversion of proton pumping direction which suggests that E205 and E215 constitute the proton accepting group (PAG) upon formation of the M intermediate. These residues correspond to Q213 and E227 in *Ta*HeR, and Q216 and E230 in 48C12, which

constitute a water-containing hydrogen-bonding network (*Kovalev et al., 2020*; *Lu et al., 2020*; *Shihoya et al., 2019*). As HeRs contain a single counterion of the Schiff base, it cannot be the proton acceptor upon deprotonation of the Schiff base. Instead, in the case of V2HeR3, the E205/E215 region appears to serve as PAG that receives the proton from the Schiff base. Interestingly, when the PAG mechanism is disrupted by mutation, the protein is converted to an outward proton pump. We found that monovalent anions affect $E_{rev}$, but only at the extracellular side (*Figure 1—figure supplement 4*), suggesting anion binding to V2HeR3 (*Figure 4D*).

The discovery of proton-transporting HeRs provides various insights regarding evolution, physiology, mechanism of action, and application of this rhodopsin family. Phylogenetic analysis strongly indicates that coccolithoviruses acquired their HeR genes from their past algal hosts (see *Figure 4—figure supplement 1*). The split into proton-transporting and non-transporting types took place early in the evolution of the viral HeRs, while their distribution among coccolithoviruses suggests that the last common ancestor of all known isolates might have already possessed genes of both types (*Figure 4—figure supplement 2*). The proton-transporting HeR was then secondarily lost in the lineage of *Eh*V-201. Proton-transporting HeRs are thus an innovation that appeared after the viruses acquired HeRs from algae, but the exact benefit that this function provides to the viruses is unclear. It was reported that light is required for viral adsorption to *E. huxleyi* cells (*Thamatrakoln et al., 2019*). If V2HeR3 is expressed in *E. huxleyi* cell membranes, light might depolarize the host cell membranes by this protein. It can be thus speculated that depolarization helps the virus to overcome cell defense or prevents superinfection. As *E. huxleyi* blooms significantly affect the marine environment and climate, the influence of light is intriguing. The viral HeRs are the first rhodopsin transporters to be characterized from the family *Phycodnaviridae*, paralleled by the type-1 rhodopsin channels in the *Mimiviridae* (*Rozenberg et al., 2020*; *Zabelskii et al., 2020*).

Type-1 microbial rhodopsins' functions are diverse, while we now know that some HeRs participate in ion-transport such as proton channeling and pumping. The functional conversion of V2HeR3 into an outward proton pump may suggest the presence of such HeR proteins in nature. Thus, HeR's functions are likely to be diverse similarly to type-1 rhodopsins. Finally, type-1 ion-transporting rhodopsins have been used as the main tools in optogenetics (*Boyden et al., 2005*; *Deisseroth and Hegemann, 2017*; *Ishizuka et al., 2006*), and ion-transporting HeRs could be potential optogenetic tools as well.

## Methods

### Sequence extraction

HeR genes from *Eh*V-202, *Eh*V-201, and *Eh*V-PS401 were obtained from the GenBank genome assemblies HQ634145.1 (*Nissimov et al., 2012a*), JF974311.1 (*Nissimov et al., 2012b*), and HQ634146.1 (Bioproject PRJNA47633), respectively: AET42597.1 (V2HeR1), AET42570.1 (V2HeR2), and AET42421.1 (V2HeR3) for *Eh*V-202, AET97940.1 (V1HeR1), AET97964.1 (V1HeR2) for *Eh*V-201, and AET73409.1 (VPS401HeR) for *Eh*V-PS401. Sequences related to the viral HeRs were searched for in the metagenomic databases Ocean Microbial Reference Catalog v.1 and v.2 (*Villar et al., 2018*), as well as other assemblies of the Tara Oceans data (*Philosof et al., 2017*; *Sunagawa et al., 2015*), with blastp v. 2.11.0+ (*Altschul et al., 1990*). After removing partial and redundant sequences, three full-length genes with a varying degree of relatedness to V2HeR3 were retained coming from contigs SAMEA2621401_1124616, SAMEA2621075_258957, and TARA_B100000767_G_C19749265_1 (containing OM-RGC gene OM-RGC.v1.010885482). The corresponding genes were dubbed V*Tara*4616HeR, V*Tara*8957HeR, and V*Tara*5482HeR, respectively. Viral origin of the contigs is supported by matches to *Eh*Vs (*Supplementary file 3*). Annotated versions of the contigs are provided in *Supplementary file 1*. The sequence used to represent the HeR gene from the host species (*Eh*HeR) was obtained from transcriptome assemblies of *E. huxleyi* str. PLY M219 (corresponding to NCBI TSA transcript HBOB01045747.1). The gene is putatively single copy, but has several splice variants and is detected in transcriptome assemblies of several *E. huxleyi* strains with minor allelic variation (*Supplementary file 3*). The gene is absent from the genome assembly of *E. huxleyi* CCMP1516 (NCBI Assembly GCA_000372725.1). Pairwise protein sequence identities are provided in *Supplementary file 3*. The identities were obtained by extracting transmembrane regions predicted for the full set of HeRs from the viruses and algae (see below) with PolyPhobius (*Käll et al., 2005*).

## Phylogenetic reconstructions

Sequences for the analysis of the relationships between the viral and the eukaryotic HeRs were collected by searching a collection of 1315 transcriptomes and genomes from algae and other unicellular eukaryotes from NCBI Assembly, NCBI TSA, MMETSP (*Johnson et al., 2019*; *Keeling et al., 2014*), 1KP (*Leebens-Mack, 2019*), reefgenomics.org (*Liew et al., 2016*), as well de novo assemblies of data from NCBI SRA. If not annotated in the source databases, genes were predicted using GeneMark-ES v. 4.62 (*Lomsadze et al., 2005*) in the genomes or using Transdecoder v. 5.5.0 (*Haas et al., 2013*) in the transcriptomes. HeRs sequences were retrieved by searching the resulting protein database with hmmsearch from HMMER v. 3.3.2 (*Eddy, 1996*) using the Pfam HeR profile PF18761.4 with an E-value threshold of 1e-5, resulting in 565 sequences. Proteins most similar to the viral HeRs were obtained by searching among the HeRs using blastp with viral HeRs as queries with an E-value threshold of 1e-10. The resulting 268 sequences were clustered at 90% protein identity using cdhit v. 4.8.1 (*Li and Godzik, 2006*), truncated and mis-annotated sequences were removed and the resulting representative sequences together with the viral HeRs were aligned with mafft v. 7.475 (`--local-pair --maxiterate 1000`) (*Katoh et al., 2002*), trimmed with trimal v. 1.4.rev15 (`-gt 0.9`) (*Capella-Gutiérrez et al., 2009*), and the phylogeny was reconstructed with iqtree v. 2.1.2 (*Minh et al., 2020*). The tree was midpoint-rooted. For a comparison to the more distant HeRs from prokaryotes and from *Micromonas commoda,* a selected set of *Eh*V sequences and related algal HeRs were taken for a phylogenetic reconstruction using the same strategy.

Phylogenetic relationships between *Coccolithovirus* isolates were reconstructed by collecting orthologous genes with proteinortho v. 6.0.25 (*Lechner et al., 2011*) with blastp as the search engine. Viruses *Ectocarpus siliculosus* virus-1 and *Feldmannia* species virus 158 from the sister genus *Phaeovirus* were recruited as outgroups. Strictly single-copy orthogroups present in at least 14 of the 16 genomes were selected and protein sequences were aligned with mafft (`--localpair --maxiterate 1000`), trimmed with trimal (`-automated1`) and the phylogeny was reconstructed with iqtree2 with partitions. Details on the HeR genes and phylogenetic markers in the used genomes are available in *Supplementary file 2*. The history of HeR gene duplications and losses among the three *Coccolithovirus* lineages was reconstructed with Notung v. 2.9.1.5 (*Chen et al., 2000*) using default costs, under the assumption of no gene transfer between lineages.

An unrooted phylogenetic tree in *Figure 1A* was constructed with MEGAX software (*Kumar et al., 2018*). The protein sequences were aligned using MUSCLE (*Edgar, 2004*). The evolutionary history was inferred using the Neighbor-Joining method (*Saitou and Nei, 1987*) with bootstrap values based on 1000 replications. Sequence data of the other rhodopsins were from the GenBank database.

## Expression plasmids for mammalian cells

The expression plasmid for V2HeR3 with epitope tags (cMyc) and/or eGFP. peGFP-P2A-V2HeR3 was created by the following procedure. A full-length, DNA sequence encoding V2HeR3 was purchased from GenScript Japan (Tokyo, Japan). The gene encoding V2HeR3 and peGFP-P2A or pCaMKIIα-eGFP-P2A vector were amplified by PCRs, and V2HeR1 was subcloned into a peGFP-P2A vector or pCaMKIIa-eGFP-P2A vector using an In-Fusion HD cloning kit (Takara Bio, Inc, Shiga, Japan) according to the manufacturer's instructions. For the immunostaining experiment, N-QKLISEEDL-C (10 amino acids, cMyc epitope tag) in the N-terminal or C-terminal of V2HeR3 was inserted in the plasmid peGFP-P2A-V2HeR3 using inverse PCR (KOD-Plus-Mutagenesis Kit, TOYOBO, Osaka, Japan). Site-directed mutagenesis was performed using a KOD-Plus-Mutagenesis Kit according to the manufacturer's instructions.

Synthesized genes encoding *Eh*HeR, V1HeR2, V2HeR2, VPS401HeR, V*Tara*5482HeR, V*Tara*4616HeR, and V*Tara*8957HeR were subcloned into a peGFP-P2A vector using an In-Fusion HD cloning kit. V1HeR2 and V2HeR2 were purchased from GenScript. *Eh*HeR, VPS401HeR, V*Tara*5482HeR, V*Tara*4616HeR, and V*Tara*8957HeR were purchased from GENEWIZ Japan (Azenta Life Sciences, Tokyo, Japan). V1HeR1 and V2HeR1 were synthesized by GENEWIZ and subcloned into peGFP-P2A vector by GENEWIZ.

All the constructs were verified by DNA sequencing (Fasmac Co., Ltd., Kanagawa, Japan). All the PCR primers used in this study were summarized in *Supplementary file 3*.

## Mammalian cell culture

The electrophysiological and cytochemistry assays of HeRs were performed on ND7/23 cells, hybrid cell lines derived from neonatal rat dorsal root ganglia neurons fused with mouse neuroblastoma (*Wood et al., 1990*). ND7/23 cells were grown on a collagen-coated coverslip in Dulbecco's modified Eagle's medium (FUJIFILM Wako Pure Chemical Corporation, Osaka, Japan) supplemented with 2.5 μM all-*trans* retinal, 5% fetal bovine serum under a 5% $CO_2$ atmosphere at 37°C. The expression plasmids were transiently transfected by using Lipofectamine 3000 (Thermo Fisher Scientific, Waltham, MA) according to the manufacturer's instructions. Electrophysiological recordings were then conducted 16–36 hr after the transfection. Successfully transfected cells were identified by eGFP fluorescence under a microscope prior to the measurements.

Cortical neurons were isolated from embryonic day 16 Wistar rats (Japan SLC, Inc, Shizuoka, Japan) using Nerve-Cells Dispersion Solutions (FUJIFILM Wako Pure Chemical Corporation) according to the manufacturer's instructions and grown in culture medium (FUJIFILM Wako Pure Chemical Corporation) under a 5% $CO_2$ atmosphere at 37°C. The expression plasmids were transiently transfected in cortical neurons calcium phosphate transfection at days in vitro (DIV) 5. Electrophysiological recordings were then conducted at DIV21-23 to neurons identified to express eGFP fluorescence under a conventional epifluorescence system.

## Electrophysiology

All experiments were carried out at room temperature (23 ± 2°C). Photocurrents and action potentials were recorded as previously described using an Axopatch 200B amplifier (Molecular Devices, Sunnyvale, CA) under a whole-cell patch-clamp configuration (*Hososhima et al., 2021*). The data were filtered at 5 kHz and sampled at 20 kHz (Digdata1550, Molecular Devices, Sunnyvale, CA) and stored in a computer (pClamp10.6, Molecular Devices). The pipette resistance was between 3–10 MΩ. All patch-clamp solutions are described in *Supplementary file 3*. The liquid junction potential was calculated and compensated by the pClamp 10.6 software.

For whole-cell patch clamp, irradiation at 470 or 530 nm was carried out using WheeLED (parts No. WLS-LED-0470-03 or WLS-LED-0530-03, Mightex, Toronto, Canada) controlled by computer software (pCLAMP10.6, Molecular Devices). The light power was directly measured at an objective lens of microscopy by a visible light-sensing thermopile (MIR-100Q, SSC Inc, Mie, Japan).

Transient photocurrent ($I_0$) corresponds to initial peak photocurrents during light illumination, peak photocurrent ($I_1$) corresponds to average value of photocurrents from 10 to 11 ms, steady-state photocurrent ($I_2$) corresponds to average value of photocurrents of the pulse-end 10 ms. All data in the text and figures are expressed as mean ± SEM and were evaluated with the Mann-Whitney *U* test for statistical significance, unless otherwise noted. It was judged as statistically insignificant when p>0.05.

## Cytochemistry

The cultured ND7/23 cells on glass coverslips were washed with PBS (NACALAI TESQUE, Inc, Kyoto, Japan). Two cell samples were fixed in 4% paraformaldehyde phosphate buffer solution (NACALAI TESQUE, INC.) for 15 min at room temperature. The cells were washed with PBS three times. One sample was permeabilized with 0.5% Triton X-100 for 15 min at room temperature, the other sample was not permeabilized. The cells were treated with a blocking buffer consisting of 3% goat serum for 60 min at room temperature. Then the cells were reacted with rabbit anti-c-Myc primary antibody (C3956; Sigma-Aldrich, St Louis, MO) at 1:500 dilution for 60 min at room temperature. The cells were washed with PBS three times before labeling with goat anti-rabbit IgG H&L Biotin (ab97049; abcam, Cambridge, UK) at 1:500 dilution for 30 min at room temperature. After that the cells were washed PBS three times before labeling with streptavidin, Alexa Fluor 594 (S32356; Thermo Fisher Scientific) at 1:200 dilution for 2 hr at room temperature. After a final wash with PBS three times, the coverslips were mounted on glass slides with ProLong Diamond Antifade Mountant (Thermo Fisher Scientific). Localization was assessed using an LSM880 confocal laser scanning microscope (Zeiss, Jena, Germany) equipped with ×63 oil-immersion objective lens (Zeiss) and a software ZEN (Zeiss). The captured images were analyzed with Fiji software (*Schindelin et al., 2012*).

## Expression plasmids for *P. pastoris* cells

For the purification of V2HeR3-WT, the gene encoding N-terminal His-tagged V2HeR3 was cloned into the EcoRI and XbaI site of pPICZB vector (Thermo Fisher Scientific). For determination of $\lambda_{max}$ of V2HeR3 WT and mutants, the S-tag (KETAAAKFERQHMDS) and thrombin recognized sequence (LVPRGS) were added between N-terminal His-tag and V2HeR3 coding region.

## Protein expression and purification by *P. pastoris* cells

The gene encoding N-terminal His-tagged V2HeR3 was cloned into the EcoRI and XbaI site of pPICZB vector (Thermo Fisher Scientific). The recombinant protein was expressed in the *P. pastoris* strain SMD1168H (*Philosof et al., 2017*) (Thermo Fisher Scientific). The cells were harvested 48–60 hr after expression was induced in BMMY medium when 10 mM of all-*trans*-retinal (Sigma-Aldrich) was supplemented in the culture to a final concentration of 30 µM. Additionally, 100% filtered methanol was added to the growth medium every 24 hr of induction to a final concentration of 0.5%. Membranes containing V2HeR3 was isolated as described elsewhere (*Yamauchi et al., 2017*) with the following modifications. Washed *P. pastoris* cells were resuspended in buffer A (7 mM $NaH_2PO_4$, 7 mM EDTA, 7 mM DTT, and 1 mM phenylmethylsulfonyl fluoride [PMSF], pH 6.5) and slowly shaken with all-*trans*-retinal (added to a final concentration of 25 µM) in the dark at room temperature for 3–4 hr in the presence of 0.5% of Westase (Takara Bio, Inc) to digest the cell wall. The cells were disrupted by the two-times passage through a high-pressure homogenizer (EmulsiFlex C3, Avestin, Inc, Canada). The supernatants were centrifuged for 30 min at 40,000× *g* in a fixed-angle rotor, and the V2HeR3 membrane pellets were resuspended in solubilization buffer (20 mM $KH_2PO_4$, 1% *n*-dodecyl-β-D-maltoside (DDM), 1 mM PMSF, pH 7.5) and stirred overnight at 4°C. The solubilization mixture was centrifuged for 30 min at 40,000× *g* in a fixed-angle rotor. The solubilized protein was incubated with Ni-NTA agarose (QIAGEN, Hilden, Germany) for several hours. The resin with bound V2HeR3 was washed with wash buffer (50 mM $KH_2PO_4$, 400 mM NaCl, 0.1% DDM, 35 mM imidazole, pH 7.5) and then treated with elution buffer (50 mM $KH_2PO_4$, 400 mM NaCl, 0.1% DDM, 250 mM imidazole, pH 7.5). The collected fractions were dialyzed against a solution containing 50 mM $KH_2PO_4$, 400 mM NaCl, 0.1% DDM at pH 7.5 to remove the imidazole.

## Ion transport assay of *P. pastoris* cells by pH electrode

The number of *P. pastoris* cells expressing rhodopsins was estimated by the apparent optical density at 660 nm ($OD_{660}$), and 7.5 ml cell culture ($OD_{660}=2$) were used for the experiment. The cells were washed with an unbuffered 100 mM NaCl solution three times, and resuspended in the same solution. The cell suspension was placed in the dark and then illuminated at $\lambda >500$ nm, by the output of a 1 kW tungsten-halogen projector lamp (Rikagaku, Japan) through a glass filter (Y-52, AGC Techno Glass, Japan). The light-induced pH changes were measured with a pH electrode (HORIBA, Ltd, Japan) (*Inoue et al., 2013*). Measurements were repeated under the same conditions with the addition of 30 µM CCCP, a protonophore molecule.

## HPLC analysis of retinal configuration

The HPLC was equipped with a silica column (6.0×150 mm; YMC-Pack SIL, YMC, Japan), a pump (PU-2080, JASCO, Japan), and a UV-visible detector (UV-2070, JASCO) (*Kawanabe et al., 2006*). The solvent was composed of 12% (v/v) ethyl acetate and 0.12% (v/v) ethanol in hexane with a flow rate of 1.0 ml/min. Retinal oxime was formed by a hydrolysis reaction with the sample in 100 µl solution at 0.1 mg/ml protein concentration and 50 µl hydroxylamine solution at 1 M at 0°C. To ensure all the protein molecules reacted completely, 300 µl of methanol was added to denature the proteins. For light-adapted V2HeR3, the sample solution was illuminated with $\lambda >500$ nm light (Y-52, AGC Techno Glass) for 1 min before denaturation and extraction. Then, the retinal oxime was extracted using hexane and 300 µl of solution was injected into the HPLC system. The molar composition of the retinal isomers was calculated from the areas of the corresponding peaks in the HPLC patterns. The assignment of the peaks was performed by comparing them with the HPLC pattern from retinal oximes of authentic all-*trans*, 13-*cis*, and 11-*cis* retinals. To estimate the experimental error, three independent measurements were carried out.

## pH titration

To investigate the pH dependence of the absorption spectra of V2HeR3, a solution containing about 6 µM protein was solubilized in 6-mix buffer (10 mM citrate, 10 mM MES, 10 mM HEPES, 10 mM

MOPS, 10 mM CHES, and 10 mM CAPS). The pH was then changed by the addition of concentrated HCl or NaOH (7). The absorption spectra were measured with a UV-visible spectrometer (V-2400PC, SHIMADZU, Japan) at each approximately 0.5 pH change.

### Laser flash photolysis

For the laser flash photolysis measurement, V2HeR3 was purified and solubilized in 0.1% DDM, 400 mM NaCl, and 50 mM $KH_2PO_4$ (pH 7.5). The absorption of the protein solution was adjusted to 0.5 (total protein concentration ~0.25 mg/ml) at $\lambda_{max}$ = 500 nm. The sample was illuminated with a beam from an OPO system (LT-2214, LOTIS TII, Minsk, Republic of Belarus) pumped by the third harmonics of a nanosecond pulsed $Nd^{3+}$-YAG laser ($\lambda$ =355 nm, LS-2134UTF, LOTIS TII) (*Shihoya et al., 2019*). The time evolution of transient absorption change was obtained by observing the intensity change of an output of an Xe arc lamp (L9289-01, Hamamatsu Photonics, Japan), monochromated by a monochromator (S-10, SOMA OPTICS, Japan) and passed through the sample, after photo-excitation by a photomultiplier tube (R10699, Hamamatsu Photonics, Japan). To increase the signal-to-noise (S/N) ratio, multiple measurements were averaged. The signals were global-fitted with a multiexponential function to obtain the lifetimes of each photointermediate.

### Low-temperature UV-visible and FTIR spectroscopy

The purified proteins of V2HeR3 were reconstituted into a mixture of POPE and POPG membranes (molar ratio = 3:1) with a protein-to-lipid molar ratio of 1:20 by removing DDM using Bio-Beads (SM-2; Bio-Rad, CA). The reconstituted samples were washed three times with 1 mM NaCl and 2 mM Tris-HCl (pH 8.0). The pellet was resuspended in the same buffer, where the concentration was adjusted to make the intensity of amide I~0.7. A 60 µl aliquot was placed onto a $BaF_2$ window and dried gently at 4°C. The films were then rehydrated with 2 µl $H_2O$ or $D_2O$, and allowed to stand at room temperature for 15 min to complete the hydration. For UV-visible spectroscopy, the sample film was hydrated with $H_2O$, and placed and cooled in an Optistat DN cryostat mounted (Oxford Instruments, Abingdon, UK) in a UV-vis spectrometer (V-550, JASCO, Japan) (*Kawanabe et al., 2007*). For FTIR spectroscopy, the sample film was hydrated with $H_2O$ or $D_2O$, and placed and cooled in an Oxford Optistat DN2 cryostat mounted in a Cary670 spectrometer (Agilent Technologies, Japan) (*Shihoya et al., 2019*). The 128 interferograms were accumulated with 2 $cm^{-1}$ spectral resolution for each measurement.

For the formation of the K intermediate, samples of V2HeR3 were illuminated with 520 nm light (interference filter) from a 1 kW tungsten-halogen projector lamp (Rikagaku) for 2 min at 100 K. The K intermediate was photo-reversed with $\lambda$ >590 nm light (R-61 cut-off filter, Toshiba, Japan) for 1 min, followed by illumination with 540 nm light. For the formation of the M intermediate, samples of V2HeR3 were illuminated with $\lambda$ >500 nm light (Y-52 cut-off filter, Toshiba) from a 1 kW halogen-tungsten lamp for 1 min at 230 K. The M intermediate was photo-reversed with the 400 nm light (interference filter) for 2 min, followed by illumination with $\lambda$ >500 nm light. To increase the S/N ratio in FTIR spectroscopy, photoconversions to the K intermediate at 100 K and to the M intermediate at 230 K were repeated eight and five times, respectively.

### Determination of $\lambda_{max}$ by hydroxylamine bleach

The $\lambda_{max}$ values of HeRs in ND7/23 or *P. pastoris* cells were determined by observing the bleaching by hydroxylamine upon light absorption. ND7/23 cells were grown as mentioned above. The expression plasmids were transiently transfected by using PEI-Max according to the manufacturer's instructions. ND7/23 cells (100 mm dish ×2) expressing rhodopsin were centrifuged and resuspended in 50 mM Tris-Cl (pH 8.0), 100 mM NaCl buffer to a final volume of 0.7 ml. Then, the mammalian cells were disrupted by ultrasonication and solubilized in 1.0% DDM. We added hydroxylamine to the sample (final concentration of 50 mM) and illuminated it for 16 min with a SOLIS-1C – High-Power LED (THOR-LABS, Japan) through a glass filter (Y-52, AGC Techno Glass) at wavelengths >500 nm. Absorption changes representing the bleaching of rhodopsins by hydroxylamine were measured, using a UV-visible spectrometer, V750 (JASCO) with an integrating sphere unit, ISV922.

*P. pastoris* membranes expressing rhodopsin were resuspended in 50 mM Tris-Cl (pH 8.0), 100 mM NaCl buffer to final concentration 5 mg/ml. Then, the membrane fraction was solubilized in 1.0% DDM. We added hydroxylamine to the sample (final concentration of 500 mM) and illuminated it for 16 min with a 1 kW tungsten-halogen projector lamp (Master HILUX-HR, Rikagaku) through a

glass filter (Y-52, AGC Techno Glass) at wavelengths >500 nm. Absorption changes representing the bleaching of rhodopsins by hydroxylamine were measured, using a UV-visible spectrometer V650 (JASCO) with an integrating sphere unit.

## Acknowledgements

This work was financially supported by grants from the Japanese Ministry of Education, Culture, Sports, Science and Technology (18H03986, 19H04959, 20K21251, 21H04969 to HK; 18K06109 to SPT; 17H03007 to KI; 20K15900 to SH); CREST, Japan Science and Technology Agency (JPMJCR1753 to HK); and Israel Science Foundation (FIRST program 3592/19 and Research Center grant 3131/20 to OB), OB holds the Louis and Lyra Richmond Chair in Life Sciences.

## Additional information

### Funding

| Funder | Grant reference number | Author |
| --- | --- | --- |
| Ministry of Education, Culture, Sports, Science and Technology | | Hideki Kandori<br>Shoko Hososhima<br>Keiichi Inoue<br>Satoshi P Tsunoda |
| Israel Science Foundation | | Oded Béjà |
| Japan Science and Technology Agency | | Hideki Kandori |

The funders had no role in study design, data collection and interpretation, or the decision to submit the work for publication.

### Author contributions

Shoko Hososhima, Ritsu Mizutori, Rei Abe-Yoshizumi, Andrey Rozenberg, Shunta Shigemura, Masae Konno, Kota Katayama, Keiichi Inoue, Satoshi P Tsunoda, Investigation, Writing – review and editing; Alina Pushkarev, Conceptualization, Investigation, Writing – review and editing; Oded Béjà, Conceptualization, Supervision, Funding acquisition, Writing – review and editing; Hideki Kandori, Conceptualization, Supervision, Funding acquisition, Writing - original draft

### Author ORCIDs

Shoko Hososhima  http://orcid.org/0000-0001-7465-2907
Rei Abe-Yoshizumi  http://orcid.org/0000-0002-7082-7949
Keiichi Inoue  http://orcid.org/0000-0002-6898-4347
Satoshi P Tsunoda  http://orcid.org/0000-0003-3636-1521
Oded Béjà  http://orcid.org/0000-0001-6629-0192
Hideki Kandori  http://orcid.org/0000-0002-4922-1344

### Decision letter and Author response

Decision letter https://doi.org/10.7554/eLife.78416.sa1
Author response https://doi.org/10.7554/eLife.78416.sa2

## Additional files

### Supplementary files

• Supplementary file 1. GenBank flat file. Annotated sequences of the metagenomic contigs containing V*Tara*5482HeR, V*Tara*8957HeR, and V*Tara*4616HeR.

• Supplementary file 2. Excel spreadsheet. List of viral genomes and phylogenetic markers used in this study. For each virus, accessions of heliorhodopsins (HeRs) genes and accessions for the indicated phylogenetic markers are provided.

• Supplementary file 3. Excel spreadsheet. Supplementary information for methods.

• MDAR checklist

## Data availability

All data needed to evaluate the conclusions are present in the paper. The datasets of the current study are available in the Dryad repository (https://doi.org/10.5061/dryad.31zcrjdpb).

The following dataset was generated:

| Author(s) | Year | Dataset title | Dataset URL | Database and Identifier |
|---|---|---|---|---|
| Kandori H | 2022 | Proton-transporting heliorhodopsins from marine giant viruses | https://dx.doi.org/10.5061/dryad.31zcrjdpb | Dryad Digital Repository, 10.5061/dryad.31zcrjdpb |

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
