## [Editor Report]

This manuscript provides the first experimental evidence that some members of the newly discovered heliorhodopsins can function as proton channels. The authors provide evidence of this transport function as well as a characterization of the photo cycle. The authors also demonstrate that these heliorhodopsin proton channels can be utilized as optogenetic tools. These findings should be of interest to a wide audience interested in membrane biophysics as well as in the development of tools for neuroscience.

---

## [Decision Letter]

**Decision letter after peer review:**

Thank you for submitting your article "Proton-transporting heliorhodopsins from marine giant viruses" for consideration by *eLife*. Your article has been reviewed by 3 peer reviewers, including Leon D Islas as Reviewing Editor and Reviewer #1, and the evaluation has been overseen by Richard Aldrich as the Senior Editor.

Essential revisions:

1. The main concern expressed by reviewers during the consultation and in the reviews is that the authors seem to only pay attention to the channel-like behavior (components I1 and I2), while the large initial component seems to suggest transporter-like behavior, and this is ignored for the most part.

2. Reviewers suggest that additional experiments are needed to clarify the nature of the three components, especially since some results, like the Erev of the proton current, do not coincide with the expected Nernst potential. This is indicative of the transport of other ions besides protons. Additional experiments are needed to distinguish between pure proton permeation or lack of selectivity among monovalent cations.

3. Please revise the manuscript along these lines and attend to the individual recommendations in the attached reviews.

*Reviewer #1 (Recommendations for the authors):*

1. In line 97: "Although I0 does not necessarily indicate ion transport.." This is a major component of the whole cell current and it is just glossed over. This is not a capacitive current since it is light-activated. The authors should provide an explanation for this current and perhaps an experimental characterization.

2. It is stated that the current increases linearly with light intensity, but in figure S2 it is clearly seen that the current saturates at high intensities.

3. It is also stated that the currents have linear voltage dependence and that it is steep. Linearity of the IVs indicates the absence of voltage dependence and any statement about the steepness of the IV has to be made with a quantitative measure (slope factor) and in comparison to other similar transport proteins.

4. Analysis of the steady current components indicates that, although the shift in reversal potential upon pH changes is in the correct direction for proton selectivity, it is far from the expected Nernst potential for protons, please explain possible non-selective transport of other cations or even anions.

5. While the provided exploration of the photocycle is in-depth, it assumes expert knowledge of a very specialized topic. The different intermediates (M, K, etc.) are not even defined in any part of the text. Please introduce these in the relevant part of the manuscript and provide context.

6. In figure 2b the units are missing (I assume it's time).

7. A good figure is needed illustrating the interactions of E105 to better understand the effects of mutations of proton transport.

8. While the mutants characterized here clearly show a contribution of protonatable residues to ion transport, it is important to know if the ion being transported is still protons or has the selectivity has been altered. All the functional mutants in figure 3 show currents with an altered reversal potential, indicating loss of proton selectivity, but what is the ion being transported?

*Reviewer #2 (Recommendations for the authors):*

I recommend this manuscript for publication after revision to resolve the following issues:

Lines 33-34: "Here we report that a viral HeR from Emiliania huxleyi virus 202 (V2HeR3) is a light-gated proton channel", and Lines 106-108: "The I-V plot under the symmetric ionic conditions on both sides of the cells without metal cation (pH 7.4) was almost linear with Erev of about +40 mV (Figure S3)."

The authors' interpretation of V2HeR3 as a light-gated channel does not seem to be justified by the data. A channel is expected to show zero current in the absence of the electrochemical gradient of permeant ions. In contrast, V2HeR3 generates a negative current at zero voltage under symmetrical ionic conditions (Figure S3). In fact, the behavior of V2HeR3 resembles that of cyanobacterial rhodopsin from Gloeobacter violaceus (GR), a weak (leaky) outwardly directed proton pump that generates a passive proton influx under a strong electrochemical gradient (see e.g. PMID: 24209850). Considering that the membrane topology of V2HeR3 is inverted, it acts as a leaky inward pump that generates a passive proton efflux under a high load.

Lines 97-99: "Although I0 does not necessarily indicate ion transport…"

I am fully aware that mechanistic interpretation of photocurrent components, especially recorded in response to pulses of continuous light, is a difficult task that requires a more detailed study than is expected of this first report. Nevertheless, the Authors should provide at least a hypothetic explanation of the nature of I0, considering that it has the greatest amplitude of all photocurrent components. Can it reflect an intramolecular transfer of the Schiff base proton to an inwardly located acceptor (e.g. E215, neutralization of which appears to decrease I0?).

Lines 188-190: "In Figure 3B, E205Q and E215Q show positive photocurrent signals at all the membrane voltages tested (−60 – +60 mV), suggesting proton conduction in the opposite direction (outward proton pump)."

Have the authors considered the possibility that these mutations might change the membrane topology of the protein, i.e., invert the orientation of the 7TM domain? It might provide an alternative explanation for the results obtained, especially for the E205Q mutant that generates fast positive currents (Figure 3B).

*Reviewer #3 (Recommendations for the authors):*

Discussion should be guided along the order of the figures. It is a bit unstructured.

The figure quality is very low (in my manuscript)! Please correct.

Also, I am not that familiar with the electrophysiological properties of ND7/23. Maybe it would be good to also add an IV profile for a non-infected cell?

---

## [Author Response]

Reviewer #1 (Recommendations for the authors):1. In line 97: "Although I0 does not necessarily indicate ion transport.." This is a major component of the whole cell current and it is just glossed over. This is not a capacitive current since it is light-activated. The authors should provide an explanation for this current and perhaps an experimental characterization.

We thank the reviewer’s opinion. As the reviewer pointed out, the substantial I_0_ is not a capacitive current. We performed additional experiments to characterize the I_0_ more carefully (Figure 1—figure supplement 6 and Figure 1—figure supplement 7). As shown in Figure 1—figure supplement 6, I-V plots of I0 under various ionic conditions suggest that it is dependent on intracellular pH. Thus, we proposed that H^+^ is transported. We also demonstrated that no other cation /anion is transported. In addition, we showed in Figure 1—figure supplement 7 that the peak time of I0 is independent from the membrane voltage. We added explanation of above in the text (Lines 156-170).

2. It is stated that the current increases linearly with light intensity, but in figure S2 it is clearly seen that the current saturates at high intensities.

We thank the reviewer for pointing out our mistake in the text. We corrected (Lines 110-111).

3. It is also stated that the currents have linear voltage dependence and that it is steep. Linearity of the IVs indicates the absence of voltage dependence and any statement about the steepness of the IV has to be made with a quantitative measure (slope factor) and in comparison to other similar transport proteins.

We added sentences on the linear I-V relation and mentioned about the absence of voltage dependency (Lines 116-117).

However, in our knowledge, it is not possible to determine a slop factor only from a linear I-V plot.

A slop factor is defined as a voltage/e-fold. For this, we have to know the maximum I (I_max_) where the current amplitude is saturated. In our I-V plot (Figure 1 middle and right), the current does not saturate in the measurable voltage (between -60 mV to +60 mV). Therefore, we could not determine the slope factor, unfortunately.

4. Analysis of the steady current components indicates that, although the shift in reversal potential upon pH changes is in the correct direction for proton selectivity, it is far from the expected Nernst potential for protons, please explain possible non-selective transport of other cations or even anions.

It is totally right that the E_rev_ shift upon pH change is smaller than that expected from Nernst potential. Δ pH=1.4 corresponds to 82 mV. But we observed only 40-60 mV shift (Figure 1 1D and Figure 1—figure supplement 3B). It is reasonable to argue non selective ion transport such as cation and anion. However, no shift of E_rev_ was observed, when Na^+^ or K^+^ were tested (Figure 1D), excluding the possibility of these cations to be transported. Anions are also not likely to be transported, as no effect was seen in Figure 1—figure supplement 4C (but some in Figure 1—figure supplement 4B). We reasoned the unexpected E_rev_ shift in pH change from the leaky pump function of V2HeR3. As the ionic current of V2HeR3 includes some amount of H^+^-pump component, the observed photocurrent was not fully followed by Nernst equation. We added explanations (Lines 135-141).

5. While the provided exploration of the photocycle is in-depth, it assumes expert knowledge of a very specialized topic. The different intermediates (M, K, etc.) are not even defined in any part of the text. Please introduce these in the relevant part of the manuscript and provide context.

We added sentences (Lines 196-199).

6. In figure 2b the units are missing (I assume it's time).

Added.

7. A good figure is needed illustrating the interactions of E105 to better understand the effects of mutations of proton transport.

Figure 4D was revised. We highlighted a negative charge in E105 to show the interaction between a positive charge of retinal Schiff-base

8. While the mutants characterized here clearly show a contribution of protonatable residues to ion transport, it is important to know if the ion being transported is still protons or has the selectivity has been altered. All the functional mutants in figure 3 show currents with an altered reversal potential, indicating loss of proton selectivity, but what is the ion being transported?

We thank for the valuable comment. If the V2HeR3 acts as a pure channel, the several mutants shown in Figure 3B could conduct other ions than H^+^. Because we propose V2HeR3 is a leaky proton pump (acting as a pump and a channel), E_rev_ shift could indicate that the contribution of pump and channel to the observed photocurrent was altered by amino acids substitutions. Thus, we tentatively conclude that all the mutants transport H^+^. Further studies on ion selectivity are under considerations.

Reviewer #2 (Recommendations for the authors):Lines 33-34: "Here we report that a viral HeR from Emiliania huxleyi virus 202 (V2HeR3) is a light-gated proton channel", and Lines 106-108: "The I-V plot under the symmetric ionic conditions on both sides of the cells without metal cation (pH 7.4) was almost linear with Erev of about +40 mV (Figure S3)."The authors' interpretation of V2HeR3 as a light-gated channel does not seem to be justified by the data. A channel is expected to show zero current in the absence of the electrochemical gradient of permeant ions. In contrast, V2HeR3 generates a negative current at zero voltage under symmetrical ionic conditions (Figure S3). In fact, the behavior of V2HeR3 resembles that of cyanobacterial rhodopsin from Gloeobacter violaceus (GR), a weak (leaky) outwardly directed proton pump that generates a passive proton influx under a strong electrochemical gradient (see e.g. PMID: 24209850). Considering that the membrane topology of V2HeR3 is inverted, it acts as a leaky inward pump that generates a passive proton efflux under a high load.

We thank the reviewer for pointing out a critical issue on ion channel. The I-V plot does not resemble a pure channel under a symmetrical condition. As the reviewer mentioned, GR (a proton pump rhodopsin) functions as a leaky outward proton pump that also generates passive proton influx. In addition, the previous study suggested that ChR2, a light gated cation channel contains outward proton pump current in the photocurrent, termed ChR2 as a leaky proton pump. Thus, we agree that V2HeR3 is a leaky inward pump. We have revised the text to explain the electrophysiological data in more detail (Lines 127-151).

Lines 97-99: "Although I0 does not necessarily indicate ion transport…"I am fully aware that mechanistic interpretation of photocurrent components, especially recorded in response to pulses of continuous light, is a difficult task that requires a more detailed study than is expected of this first report. Nevertheless, the Authors should provide at least a hypothetic explanation of the nature of I0, considering that it has the greatest amplitude of all photocurrent components. Can it reflect an intramolecular transfer of the Schiff base proton to an inwardly located acceptor (e.g. E215, neutralization of which appears to decrease I0?).

We thank for a valuable comment on I_o_ component which actually exhibited the largest amplitude in the photocurrent recording. We performed additional experiments to characterize the I_0_ more carefully (Figure 1—figure supplement 6 and Figure 1—figure supplement 7). As shown in Figure 1—figure supplement 6, I-V plots of I_0_ under various ionic conditions suggest that is dependent on intracellular pH. Thus we proposed that H^+^ is transported in I_0_. We also demonstrated that no other cation/anion is transported. In addition, we showed in Figure 1—figure supplement 7 that the peak time of I_0_ is independent from the membrane voltage. We added explanation of above in the text (Lines 156-170).

In addition, taken the reviewers words, it could be also possible that the I_0_ reflects an intramolecular proton transfer from the retinal Schiff-base to an acceptor residue. We added sentences concerning this point with additional references. (Lines 171-176).

Lines 188-190: "In Figure 3B, E205Q and E215Q show positive photocurrent signals at all the membrane voltages tested (−60 – +60 mV), suggesting proton conduction in the opposite direction (outward proton pump)."Have the authors considered the possibility that these mutations might change the membrane topology of the protein, i.e., invert the orientation of the 7TM domain? It might provide an alternative explanation for the results obtained, especially for the E205Q mutant that generates fast positive currents (Figure 3B).

We thank for the comment. But we do not consider a topology inversion. Topology inversion of ChR2 was reported in 2018 (Brown et al., 2018 Cell 175(4):1131-1140.e11. doi: 10.1016/j.cell.2018.09.026). They succeeded the inversion by adding a transmembrane helix (called Nx1B-TM) at the N-term of ChR2. However, it is unlikely that the single amino acid substitution e.g. E205Q or E215Q would change the topology of the whole proteins.

Reviewer #3 (Recommendations for the authors):Discussion should be guided along the order of the figures. It is a bit unstructured.

We thank the reviewer for the suggestion. Firstly, we like to describe the proton transport properties of V2HeR3 in comparison with the known ion transporting rhodopsins. Secondly, we discuss concerning possible physiological role of V2HeR3 in conjunction with the evolution. Although it seems rather unstructured, we leave the discussion as it is.

The figure quality is very low (in my manuscript)! Please correct.

We improved the Figure quality.

Also, I am not that familiar with the electrophysiological properties of ND7/23. Maybe it would be good to also add an IV profile for a non-infected cell?

ND7/23 cells are hybrid cell lines derived from neonatal rat dorsal root ganglia neurons fused with mouse neuroblastoma (Wood *et al.*, 1990). ND7/23 cells are practically used for conventional patch clamp experiments after heterologous expression of genes of interest, similarly to HEK293 cells. Please refer following articles. Thus, we think that adding I-V profile for non-transfected cell is not necessary.

1. Grimm C, Silapetere A, Vogt A, Bernal Sierra YA, Hegemann P. “Electrical properties, substrate specificity and optogenetic potential of the engineered light-driven sodium pump eKR2.” Sci Rep. 2018, 8(1):9316

2. Broser M, Spreen A, Konold PE, Peter E, Adam S, Borin V, Schapiro I, Seifert R, Kennis JTM, Bernal Sierra YA, and Hegemann P. “NeoR, a near-infrared absorbing rhodopsin.”

Nat Commun. 2020, 10;11(1):5682.

3. Bothe SN, and Lampert A.“The insecticide deltamethrin enhances sodium channel slow inactivation of human Nav1.9, Nav1.8 and Nav1.7.” Toxicol Appl Pharmacol. 2021, 428:115676. doi: 10.1016/j.taap.2021.115676.